# UNDERSTANDING WEIGHT-MAGNITUDE HYPERPARAMETERS IN TRAINING BINARY NETWORKS

**Joris Quist**[1]**, Yunqiang Li**[1,2]**, Jan van Gemert**[1]
1. Computer Vision Lab, Delft University of technology;  2. Axelera AI

## ABSTRACT

Binary Neural Networks (BNNs) are compact and efficient by using binary weights instead of real-valued weights. Current BNNs use latent real-valued weights during training, where hyper-parameters are inherited from real-valued networks. The interpretation of several of these hyperparameters is based on the magnitude of the real-valued weights. For BNNs, however, the magnitude of binary weights is not meaningful, and thus it is unclear what these hyperparameters actually do. One example is weight-decay, which aims to keep the magnitude of real-valued weights small. Other examples are latent weight initialization, the learning rate, and learning rate decay, which influence the magnitude of the real-valued weights. The magnitude is interpretable for real-valued weights, but loses its meaning for binary weights. In this paper we offer a new interpretation of these magnitude-based hyperparameters based on higher-order gradient filtering during network optimization. Our analysis makes it possible to understand how magnitude-based hyperparameters influence the training of binary networks which allows for new optimization filters specifically designed for binary neural networks that are independent of their real-valued interpretation. Moreover, our improved understanding reduces the number of hyperparameters, which in turn eases the hyperparameter tuning effort which may lead to better hyperparameter values for improved accuracy. Code is available at `https://github.com/jorisquist/Understanding-WM-HP-in-BNNs`

## 1 INTRODUCTION

A Binary Neural Network (BNN) weight is a single bit: −1 or +1, which are compact and efficient, enabling applications on, for example, edge devices. Yet, training BNNs using gradient decent is difficult because of the discrete binary values. Thus, BNNs are often (Kim et al., 2021b; Liu et al., 2020; Martinez et al., 2020) optimized with so called 'latent', real-valued weights, which are discretised to −1 or +1 by, *e.g.*, taking the positive or negative sign of the real value.

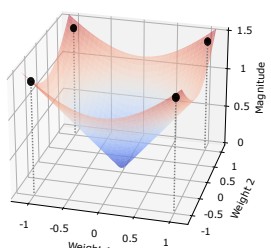

The latent weight optimization depends on several essential hyperparameters, such as their initialization, learning rate, learning rate decay, and weight decay. These hyperparameters are important for BNNs, as shown for example in Martinez et al. (2020), and also by Liu et al. (2021a), who both improve BNN accuracy by better tuning these hyperparameters.

Figure 1: Changes in real-valued weights change their magnitude. For binary weights, however, the magnitude will never change and magnitude-based hyperparameters need reinterpretation.

In this paper we investigate the latent weight hyperparameters used in a BNN, including initialization, learning rate, learning rate decay, and weight decay. All these hyperparameters influence the magnitude of the latent weights. Yet, as illustrated in Fig 1, in a BNN, the *binary weights are −1 or +1, which always have a constant magnitude and thus magnitude-based hyperparameters lose their meaning*. We draw inspiration from the seminal work of Helwegen et al. (2019), who reinterpret latent weights from an inertia perspective and state that latent weights do not exist. Thus, the

magnitude of latent weights also does not exist. Here, we investigate what latent weight-magnitude hyperparameters mean for a BNN, how they relate to each other, and what justification they have. We provide a gradient-filtering perspective on latent weight hyperparameters which main benefit is a simplified setting: fewer hyperparameters to tune, achieving similar accuracy as current, more complex methods.

## 2 RELATED WORK

**Latent weights in BNNs.** By tying each binary weight to a latent real-valued weight, continuous optimization approaches can be used to optimize binary weights. Some methods minimize the quantization error between a latent weight and its binary variant (Rastegari et al., 2016; Bulat & Tzimiropoulos, 2019). Others focus on gradient approximation (Liu et al., 2018; Lee et al., 2021; Zhang et al., 2022), or on reviving dead weights (Xu et al., 2021; Liu et al., 2021b), or on entropy regularization (Li et al., 2022) or a loss-aware binarization (Hou et al., 2017; Kim et al., 2021a). These works directly apply traditional optimization techniques inspired by real-valued network such as weight decay, learning rate and its decay, and optimizers. The summary of De Putter & Corporaal (2022) gives a good overview of these training techniques in BNNs. Recently, some papers (Liu et al., 2021a; Martinez et al., 2020; Hu et al., 2022; Tang et al., 2017) noticed that the interpretation of these optimization techniques does not align with the binary weights of BNNs (Lin et al., 2017; 2020). Here, we aim to shed light on why, by explicitly analyzing latent weight-magnitude hyperparameters in a BNN.

**Latent weight magnitudes.** Several techniques exploit the magnitude of the latent weights during BNN optimization. Latent weights clipping is proposed in (Courbariaux et al., 2015) and followed by its extensions (Alizadeh et al., 2018; Hubara et al., 2016) to clip the latent weights within a $[-1, 1]$ interval to prevent the magnitude of latent weights from growing too large. Gradient clipping (Cai et al., 2017; Courbariaux et al., 2015; Qin et al., 2020) stops gradient flow if the magnitude of latent weight is too large. Work on latent weight scaling (Chen et al., 2021; Qin et al., 2020) standardizes the latent weights to a pre-defined magnitude. Excellent results are achieved by a two-step training strategy (Liu et al., 2021a; 2020) that in the first step trains the network from scratch using only binarizing activations with weight decay, and then in the second step they fine-tune by training without weight decay. Our method reinterprets the meaning of the magnitude based weight decay hyperparameter in optimizing BNNs from a gradient filtering perspective, offering similar accuracy as two step training with a simpler setting, using just a single step.

**Optimization by gradient filtering.** Gradient filtering is a common approach used to tackle the noisy gradient updates caused by minibatch sampling. Seminal algorithms including Momentum (Sutskever et al., 2013) and Adam (Kingma & Ba, 2015) which use a first order infinite impulse response filter (IIR), *i.e.* exponential moving average (EMA) to smooth noisy gradients. Yang (2020) takes this one step further and introduces the Filter Gradient descent Framework that can use different types of filters on the noisy gradients to make a better estimation of the true gradient. In binary network optimization, Bop (Helwegen et al., 2019) and its extension (Suarez-Ramirez et al., 2021) introduce a threshold to compare with the smoothed gradient by EMA to determine whether to flip a binary weight. In our paper, we build on second order gradient filtering techniques to reinterpret the hyperparameters that influence the latent weight updates.

**Sound optimization approaches.** Instead of using heuristics to approximate gradient descent on discrete binary values, several works take a more principled approach. Peters & Welling (2018) propose a probabilistic training method for BNN, and Shekhovtsov & Yanush (2021) present a theoretical understanding of straight through estimators (STE) (Bengio et al., 2013). Meng et al. (2020) propose a Bayesian perspective and Louizos et al. (2018) formulate a noisy quantizer. Even though these approaches provide more theoretical justification in optimizing BNNs, they are more complex by either relying on stochastic settings or discrete relaxation training procedures. Moreover, these methods do not (yet) empirically reach a similar accuracy as current mainstream heuristic methods (Liu et al., 2018; 2020). In our paper, we build on the mainstream approaches, to get good empirical results, but add a better understanding of their properties, taking a step towards better theoretical understanding of empirical approaches.

## 3  HYPERPARAMETER ANALYSIS THROUGH GRADIENT FILTERING

We start with a latent weights BNN and convert it to an equivalent latent-weight free setting, as in Helwegen et al. (2019). To do this, we use a magnitude independent setting, which means that no gradient-clipping or scaling based on the channel-wise mean of the latent-weights is used.

**BNN setup.**  We use Stochastic Gradient Descent (SGD) with weight decay and momentum as a starting point, as this is a commonly used setting, see Rastegari et al. (2016), Liu et al. (2018), Qin et al. (2020). Our setup is as follows:

$$w_0 = \text{init}() \tag{1} \qquad m_i = (1 - \gamma)m_{i-1} + \gamma \nabla_{\theta_i}, \tag{2}$$

$$w_i = w_{i-1} - \epsilon(m_i + \lambda w_{i-1}) \tag{3} \qquad \theta_i = \text{sign}(w_i) \tag{4}$$

$$\text{sign}(x) = \begin{cases} -1, & \text{if } x < 0; \\ +1, & \text{if } x > 0; \\ \text{random}\{-1, +1\} & \text{otherwise.} \end{cases} \tag{5}$$

Here, $w_i$ is a latent weight at iteration $i$ which is initialized at $w_0$. $\theta_i$ is a binary weight, $\epsilon$ is the learning rate, $\lambda$ is the weight decay factor, $m_i$ is the momentum exponentially weighted moving average with $m_{-1} = 0$ and discount factor $\gamma$, $\nabla_{\theta_i}$ is the gradient over the binary weight and random$\{-1, +1\}$ is a uniformly randomly sampled -1 or +1.

We then convert to the latent-weight free setting of Helwegen et al. (2019) where latent weights are interpreted as accumulating negative gradients. We introduce $g_i = -w_i$, which allows working with gradients instead of with latent weights. We can then write Eq 3 as follows

$$g_i = g_{i-1} + \epsilon(m_i - \lambda g_{i-1}). \tag{6}$$

**Latent weight initialization.**  To investigate latent weight initialization we unroll the the recursion in Eq 6 by writing it out as a summation:

$$g_i = (1 - \epsilon\lambda)g_{i-1} + \epsilon m_i \quad = \quad \epsilon \sum_{r=0}^{i} (1 - \epsilon\lambda)^{i-r} m_r. \tag{7}$$

Latent-weights are typically initialized using real-valued weight initialization techniques (Glorot & Bengio, 2010; He et al., 2015). However, since we now interpret latent weights as accumulated gradients, we argue to also initialize them as gradient accumulation techniques such as Momentum (Sutskever et al., 2013) and simply initialize $w_0 = g_0 = 0$, because at initialization there is no preference for negative or positive gradients, and their expectation is 0. We do not use a bias-correction as done in (Kingma & Ba, 2015) because in practice we noticed that gradient magnitudes are large in the first few iterations. Applying bias correction increases this effect, which had a negative effect on training. To prevent all binary weights $\theta$ to start at the same value, we use the stochastic sign function in Eq 5 that randomly chooses a sign when the input is exactly 0.

**Learning rate and weight decay.**  The learning rate $\epsilon$ appears in two places in Eq 7: once outside the summation, and once inside the summation. The $\epsilon$ outside the summation can only scale the latent weight and will not influence outcome of the sign in Eq 4 as

$$\text{sign}\left(\epsilon \sum_{r=0}^{i} (1 - \epsilon\lambda)^{i-r} m_r\right) = \text{sign}\left(\sum_{r=0}^{i} (1 - \epsilon\lambda)^{i-r} m_r\right). \tag{8}$$

Thus, the leftmost $\epsilon$ can be removed, or set randomly without influencing the training process.

For the $\epsilon$ inside the summation of Eq 7, it appears together with the weight decay term $\lambda$. Thus, there are two free hyperparameters that only control one factor, therefore one of them is redundant and can use a single combined hyperparameter $\alpha = \epsilon\lambda$. Instead of setting a value for the learning rate $\epsilon$, and setting a value for the weight decay $\lambda$, we now only have to set a single value for $\alpha$. Since Eq 8 shows us that we can freely scale the sum with any constant factor, we scale it with $\alpha$, as

$$g_i = \alpha \sum_{r=0}^{i} (1 - \alpha)^{i-r} m_r, \tag{9}$$

which allows us to re-write the sum with a recursion, as an exponential moving average (EMA) as

$$g_i = (1 - \alpha)g_{i-1} + \alpha m_i, \tag{10}$$

where $g_{-1} = 0$. This shows that for BNNs under magnitude independent conditions, SGD with weight decay is just a exponential moving average. This gives a magnitude-free justification for using weight decay since its actual role is to act as the discount factor in an EMA. Note that it is no longer possible to set $\alpha$ to 0 since then there are no updates anymore, but setting to a small ($10^{-20}$) number will essentially work the same. The meaning of $\alpha$ is now clear, as in the EMA it controls how much to take the past into account.

**Learning rate decay.**   There no longer is a learning rate to be decayed, however, since learning rate decay scales the learning rate and $\alpha = \epsilon\lambda$, now the learning rate decay directly scales $\alpha$, so from now on we apply it to alpha and will refer to it as $\alpha$-decay. This also helps better explain its function: $\alpha$-decay increases the window size during training, causing the filtered gradient to become more stable and allowing the training to converge.

**Momentum.**   Now adding back the momentum term of Eq 2 in the original setup yields

$$m_i = (1 - \gamma)m_{i-1} + \gamma\nabla_{\theta_i}, \tag{11}$$
$$g_i = (1 - \alpha)g_{i-1} + \alpha m_i, \tag{12}$$
$$\theta_i = -\text{sign}(g_i). \tag{13}$$

Thus, SGD with weight decay and momentum is smoothing the gradient twice with an EMA filter.

**Latent weight optimization as a second order linear infinite impulse response filter.**   EMAs are a specific type of linear Infinite Impulse Response (IIR) Filter (Proakis, 2001). Linear filters are filters that compute an output based on a linear combination of current and past inputs and past outputs. The general definition is given as a difference equation:

$$y_i = \frac{1}{a_0}(b_0 x_i + b_1 x_{i-1} + ... + b_P x_{i-P} - a_1 y_{i-1} - a_2 y_{i-2} - ... - a_P y_{i-Q}), \tag{14}$$

where $i$ is the time step, $y_i$ are the outputs, $x_i$ are the inputs, $a_j$ and $b_j$ are the filter coefficients and $P$ and $Q$ are the maximum of iterations the filter looks back at the inputs and outputs to compute the current output. The maximum of $P$ and $Q$ defines the order of the filter. An EMA only looks at the previous output and the current input, so is therefore a first order IIR filter. Expressing an EMA as a filter looks as follows:

$$y_i = (1 - \alpha)y_{i-1} + \alpha x_i = \frac{1}{a_0}(b_0 x_i - b_1 \cdot x_{i-1} - a_1 y_{i-1}), \quad b = \begin{bmatrix} \alpha \\ 0 \end{bmatrix}, a = \begin{bmatrix} 1 \\ \alpha - 1 \end{bmatrix}. \tag{15}$$

In our optimizer we have a cascade of two EMAs applied in series to the same signal which can be represented by a filter with the order being the sum of the orders of the original filters. To get the new $a$ and $b$ vectors the original ones are convolved with each other. In our case this gives:

$$b = \begin{bmatrix} \gamma \\ 0 \end{bmatrix} \star \begin{bmatrix} \alpha \\ 0 \end{bmatrix} = \begin{bmatrix} \alpha\gamma \\ 0 \\ 0 \end{bmatrix}, \quad a = \begin{bmatrix} 1 \\ \gamma - 1 \end{bmatrix} \star \begin{bmatrix} 1 \\ \alpha - 1 \end{bmatrix} = \begin{bmatrix} 1 \\ (\alpha - 1) + (\gamma - 1) \\ (\alpha - 1) \cdot (\gamma - 1) \end{bmatrix}, \tag{16}$$

when applied to our gradient filtering setting in Eq 12 gives the difference equation:

$$g_i = \alpha\gamma\nabla_{\theta_i} - (\alpha + \gamma - 2)g_{i-1} - (\alpha - 1)(\gamma - 1)g_{i-2} \tag{17}$$

Thus, in a magnitude independent setting, SGD with weight decay and momentum is equivalent to a 2nd order linear IIR filter. Note that $\alpha$ and $\gamma$ have the same function: Without $\alpha$ decay, the values for $\alpha$ and $\gamma$ can be swapped. This filtering perspective opens up new methods of analysis for optimizers.

**Main takeaway.** Our re-interpretations reduces the 7 hyper parameters in the latent weight view with SGD, to only 3 hyperparameters in our filtering view, see Table 1.

|  | Learning rate | Learning rate decay | Init | Momentum | Weight decay | Scaling | Clipping |
|---|---|---|---|---|---|---|---|
| Latent: | $\epsilon$ | $\epsilon$-decay | $w_0$ | $\gamma$ | $\lambda$ | ✓ | ✓ |
| Filtered: | − | $\alpha$-decay | − | $\gamma$ | $\alpha$ | − | − |

Table 1: Hyperparameters used in the latent weight view versus our filtered gradients perspective. Our filtered perspective reduces the number of hyperparameters from 7 to 3.

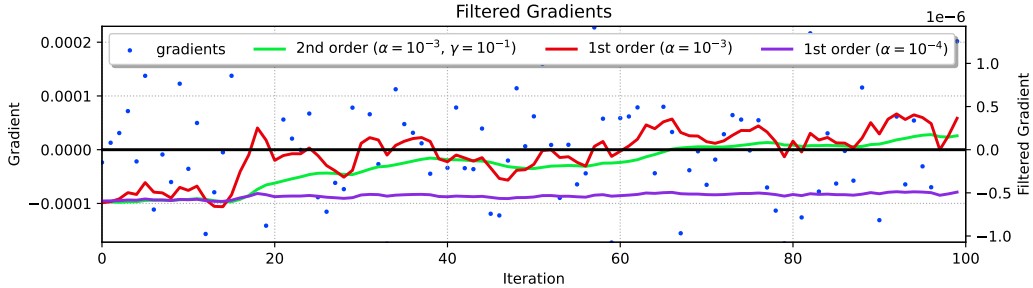

Figure 2: Gradients and filtered gradients for a first and second order filter in a single epoch on CIFAR-10. For better visualisation, the filter outputs are scaled up to a similar range as the unfiltered gradients. It can be seen that the unfiltered gradients are noisy and that the filtered outputs are smoother. The second order filter reduces the noise even further compared to the first order filter.

## 4 EXPERIMENTS

We empirically validate our analysis on CIFAR-10, using the BiRealNet-20 architecture (Liu et al., 2018). Unless mentioned otherwise the networks were optimized using SGD for both the real-valued and binary parameters with as hyperparameters: learning rate=0.1, momentum with $\gamma = (1 - 0.9)$, weight decay=$10^{-4}$, batch size=256 and cosine learning rate decay and cosine alpha decay. We analyze the weight flip ratio at every update, which is also known as the FF ratio (Liu et al. (2021a)).

$$\mathbf{I}_{\text{FF}} = \frac{|\text{sign}(w_{i+1}) - \text{sign}(w_i)|_1}{2} \qquad \mathbf{FF}_{\text{ratio}} = \frac{\sum_{l=1}^{L} \sum_{w \in W_l} \mathbf{I}_{\text{FF}}}{N_{\text{total}}} \qquad (18)$$

where $w_i$ is a latent weight at time $i$, $L$ the number of layers, $W_l$ the weights in layer $l$, and $N_{\text{total}}$ the total number of weights.

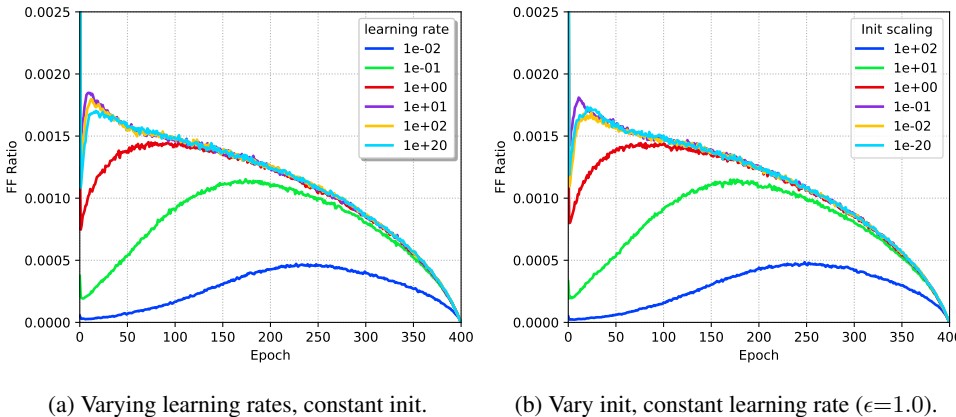

(a) Varying learning rates, constant init.

(b) Vary init, constant learning rate ($\epsilon$=1.0).

Figure 3: In the magnitude independent setting, scaling the learning rate has the exact same effect on the flipping ratio as scaling the initial latent-weights by the inverse.

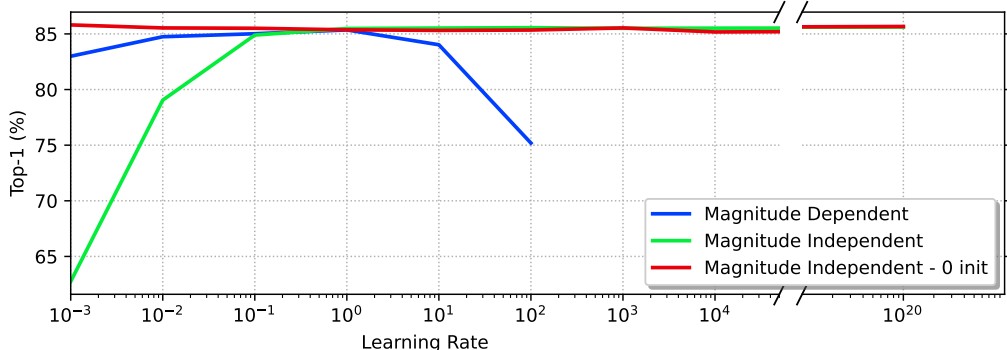

Figure 4: Learning rate effect on accuracy for three settings. (a). standard SGD. (b) Magnitude independent in Eq 3, by removing clipping and scaling. (c) Magnitude independent with latent weight initializes of 0 in Eq 7. Setting (a) has just a single optimum. The accuracy in setting (b) is sensitive to small learning rates. For setting (c), accuracy is independent of the learning rate.

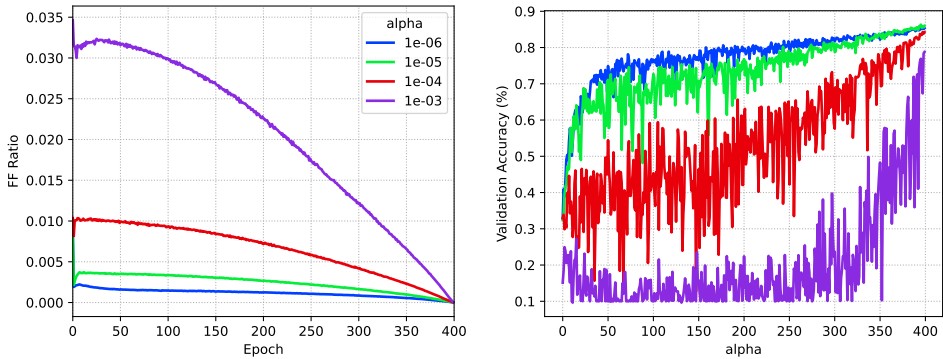

Figure 5: Bit flipping (FF) ratio and accuracy for varying alphas from eq 17. BNNs are sensitive to alpha, similar to how they are sensitive to weight decay. Tuning alpha is essential.

**1st order vs 2nd order** We visually compare filter orders by sampling real gradients from a single binary weight trained on CIFAR-10 in Figure 2. For the same $\alpha$, a 1st order filter is more noisy than a 2nd order filter. This may cause the binary weight to oscillate, even though the larger trend is that it should just flip once. To reduce these oscillations with a 1st order filter requires a smaller alpha. This, however, causes other problems because $\alpha$ determines the window size of past gradients and with a smaller $\alpha$ many more gradients are used. This means that it takes much longer for a trend in the gradients to effect the binary weight. Instead, the 2nd order filter has both benefits: it can filter out high frequency noise while still able to react quicker to changing trends.

**Magnitude independent learning rate vs initialization** In Figure 3 we show the bit flipping ratio for the learning rate $\epsilon$ and the initialization $g_0$. Multiplying $\epsilon$ with some scaling factor $s$ is the same as dividing $g_0$ by $s$: $\text{sign}\left((1-\epsilon\lambda)^i g_0 + s\epsilon \sum_{r=1}^{i}(1-\epsilon\lambda)^{i-r}m_r\right) = \text{sign}\left((1-\epsilon\lambda)^i \frac{g_0}{s} + \epsilon \sum_{r=1}^{i}(1-\epsilon\lambda)^{i-r}m_r\right)$, because the magnitude has no effect on the sign. Larger $\epsilon$ in Figure 3(a) and smaller $g_0$ in Figure 3(b) are independent to scaling and have similar flipping ratios. A too small $\epsilon$ or too large $g_0$ do not reach the same flipping ratios, because their ratio is insufficient to update the binary weights. For sufficiently large ratios it means that scaling both $\epsilon$ and $g_0$ has no effect on training, but also that scaling the $\epsilon$ or scaling $g_0$ with the inverse is identical: as seen by comparing the two plots in Figure 3.

**Sensitivity to hyperparameters for weight magnitude (in)dependence** We evaluate SGD in the standard magnitude dependent setting with clipping and scaling vs a magnitude independent setting with initializing the latent-weights to zero. To keep the effect of weight decay constant, we scale the weight decay factor inversely with the learning rate. Results in Figure 4 show that for the standard magnitude dependent setting the learning rate $\epsilon$ has to be carefully balanced. A too small $\epsilon$ w.r.t. to the ini-

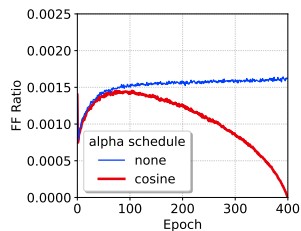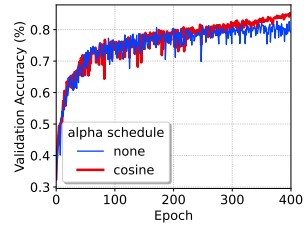

Figure 6: Flipping rate and accuracy for alpha decay. With decay the FF ratio goes to zero. Without decay, the flipping will continue, preventing convergence, reducing accuracy.

tial weights inhibits learning; while a too large $\epsilon$ will push latent-weights to the clipping region and will stop updating. In the magnitude independent setting there is no clipping. A too small $\epsilon$, however is still problematic because the magnitudes of the gradients are smaller when not using the scaling factor and the accuracy drops significantly. When initializing to zero, as we propose, this problem disappears, because there is no initial weight to hinder training and all learning rates perform equal.

**Alpha** In Fig 5 we vary $\alpha$ from eq 17. Alpha strongly influences training. A too large $\alpha$ causes too many binary weight flips per update, which hinders converging. A too small $\alpha$ makes the network converge too quickly, which hurts the end result. Tuning $\alpha$ is essential.

**Alpha decay** For proper convergence the flipping (FF) ratio should go to zero. We transform learning rate decay to alpha decay. When $\alpha$ becomes smaller, the gradients will change less, causing fewer flips, forcing the network to converge.

See the plots in Figure 6 where one network has been trained with cosine alpha decay and one without alpha decay. With and without alpha decay both seem to perform well at the start of training, however, the variant without alpha decay plateaus at the end of

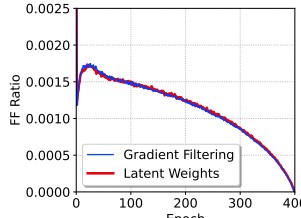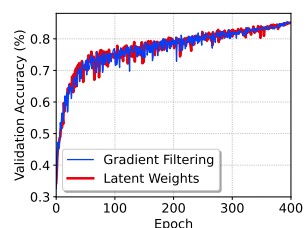

Figure 7: Equivalence of latent weights (Eq 3) and our gradient filtering (Eq 17). They are empirically equivalent.

training while the BNN with alpha decay converges better and continues improving, leading to a better end result.

**Equivalent interpretation** Figure 7 shows empirical validation with matching hyperparameters that the SGD setting using latent weights in Eq 3 is equivalent to our gradient filtering interpretation in Eq 17 that no longer uses latent weights,

### 4.1 VALIDATION OF EQUIVALENCE TO THE CURRENT STATE OF THE ART

We validate on for CIFAR-10 and Imagenet that our filtering-based optimizer is similar to the current state of the art. Several current methods use an expensive two-step optimization step. We aim to show the value of our re-interpretation by showing similar accuracy but only in a single step.

**CIFAR-10:** We train all networks for 400 epochs. As data augmentation we use padding of 4 pixels, followed by a 32x32 crop and random horizontal flip. We use Bi-RealNet-20, and for the real-valued parameters and latent-weights when used, we use SGD with a learning rate of 0.1 with cosine decay, momentum of 0.9 and on the non-BN parameters a weight decay of $10^{-4}$. For our filtering-based optimizer we used an alpha of $10^{-3}$ with cosine decay and a gamma of $10^{-1}$. Results in Table 2 show that our re-interpretation achieves similar accuracy.

Table 2: State-of-the-art on CIFAR-10. The $\star$ denotes that we re-ran these experiments ourselves.

| Method | Training Strategy | Bit-width (W/A) | Top-1 Acc(%) |
|---|---|---|---|
| FP | | 32/32 | 91.7 |
| DoReFa-Net (Zhou et al., 2016) | | 1/1 | 79.3 |
| DSQ (Gong et al., 2019) | | 1/1 | 84.1 |
| IR-Net (Qin et al., 2020) | One step | 1/1 | 86.5 |
| Bi-Real$^\star$ (Liu et al., 2018) | | 1/1 | 85.0 |
| Bi-Real + Our filtering optimizer | | 1/1 | 86.5 |
| Bi-Real$^\star$ (Liu et al., 2018) | Two step | 1/1 | 86.7 |

Table 3: Comparison with state-of-the-art on Imagenet.

| Method | Training Strategy | Top-1 Acc(%) | Top-5 Acc(%) |
|---|---|---|---|
| CI-BCNN (Wang et al., 2019) | | 59.9 | 84.2 |
| Binary MobileNet (Phan et al., 2020b) | | 60.9 | 82.6 |
| MoBiNet (Phan et al., 2020a) | | 54.4 | 77.5 |
| EL (Hu et al., 2022) | One step | 56.4 | – |
| MeliusNet29 (Bethge et al., 2020) | | 65.8 | – |
| ReActNet-A + Our filtering optimizer | | **69.7** | **88.9** |
| StrongBaseline (Martinez et al., 2020) | | 60.9 | 83.0 |
| Real-to-Binary (Martinez et al., 2020) | Two step | 65.4 | 86.2 |
| ReActNet-A (Liu et al., 2020) | | 69.4 | 88.6 |
| ReActNet-A-AdamBNN (Liu et al., 2021a) | | **70.5** | **89.1** |

**Imagenet:** We follow Liu et al. (2021a): We train for 600K iterations with a batch size of 510. For the real-valued parameters we use Adam with a learning rate of 0.0025 with linear learning rate decay. For the binary parameters we use our 2nd order filtering optimizer with $\alpha = 10^{-5}$, which we decay linearly and $\gamma = 10^{-1}$. We do not use two-step training to pre-train the latent-weights. Results in Table 3, show that ReActNet-A with our optimizer compares well to other one step training methods. It approaches the accuracy of two step training approaches, albeit without an additional expensive second step of training.

## 4.2 Empirical advantage of having fewer hyperparameter to tune

Our filtering perspective significantly reduces the number of hyperparameters (Table 1). Here, we empirically verify the computational benefit of having fewer hyperparameters to tune when applied in a setting where the hyperparameters are unknown. To show generalization to other architectures and modality we use an audio classification task (Becker et al., 2018) with a fully connected network. Specifically, we use 4 layers with batch normalization of which the first layer is not binarized. For the latent weights we optimize 6 hyperparameters, and for our filtering perspective we optimize two hyperparameters, see Table 1. We did not tune learning rate decay as this had no effect on both methods. To fairly compare hyperparameter search we used Bayesian optimization (Balandat et al., 2020). The results for 25 trials of tuning, averaged over 20 independent runs are in Figure 8. We confirm that both perspectives achieve similar accuracy when their hyperparameters are well tuned. Yet, for the latent weights, it takes on average around 10 more trials when compared to the gradient filtering. This means that the latent weights would have to train many more networks, which on medium-large datasets such as Imagenet would already take several days to converge. In contrast, the gradient filtering requires much less time and energy to find a good model.

## 5 Discussion and limitations

One limitation of our work is that we do not achieve "superior performance" in terms of accuracy. Our approach merely matches the state of the art results. Note, however, that our goal is to provide insight into how SGD and its hyperparameters behave, not to improve accuracy. Our analysis ended up with an optimizer with less hyperparameters, that also have a better explanation in the context of BNN optimization leading to simpler, more elegant methods. Our main empirical contribution is in the significant computational reduction in hyperparameter tuning.

Another perceived limitation is that our new proposed optimizer can be projected back to a specific setting within the current SGD with latent-weights interpretation. Thus, our analysis might not be needed. While it is true that latent-weights can also be used, we argue that there is no disadvantage to switching to the filtering perspective, because the options are the same, but the benefit is that our hyperparameters make more sense. The option to project back to latent-weights also works the other way around and for those who already have a well tuned SGD optimizer could use it to make it easier to switch to our filtering perspective. The benefit of our interpretations is having fewer hyperparameters to set.

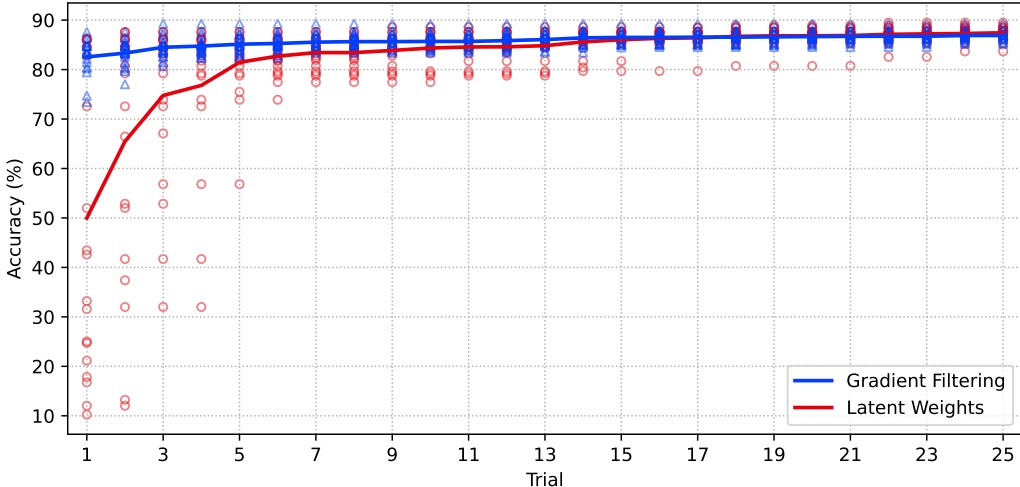

Figure 8: Bayesian hyperparameter optimization for our gradient filtering vs. latent weights. We optimize the hyperparameters of Table 1, except learning rate decay and alpha decay, as they had no influence. The scatter plot show the best achieved result so far at each trial for all runs. The lines show the average best result, up to the current trial. Both methods perform equally for good hyperparameters. Our gradient filtering view needs much less trials.

Its also true that our method cannot use common techniques based on the magnitude such as weight clipping or gradient clipping. Yet, we do not really think these techniques are necessary. We see such methods as heuristics to reduce the bit flipping ratio over time, which helps with convergence. However, in our setting, this can also be done using a good $\alpha$ decay schedule without reverting to such heuristics, making the optimization less complex.

We did not yet have the opportunity to test the filtering-based optimizer on more architectures and datasets. However, since our optimizer is equivalent to a specific setting of SGD, we would argue that architectures that have been trained with SGD will probably also work well with our optimizer. This is also a reason why we chose to use ReActNet-A, since it was trained using Adam in both in the original paper (Liu et al., 2020) and in Liu et al. (2021a). The latter specifically argues that Adam works better for optimizing BNNs, but we suspect that the advantages of Adam are decreased because it might not work in the same way in the magnitude invariant setting, as we see a smaller difference in accuracy. Introducing this normalizing aspect into the filtering-based perspective is an interesting topic for future research.

One last point to touch upon is soundness. Even though the filtering perspective provides a better explanation to hyperparameters, it does not provide understanding on why optimizing BNNs with second-order low pass filters works as well as it does. Whereas stochastic gradient descent has extensive theoretical background, this does not exist for current mainstream BNN methods. Fully understanding BNN optimization is an interesting direction for future research and our hope is that this work takes a step in that direction.

**Ethics Statement** We believe that this research does not bring up major new potential ethical concerns. Our work makes training BNNs easier, which might increase their use in practice.

**Reproducibility Statement** All our code is availabe online: `https://github.com/jorisquist/Understanding-WM-HP-in-BNNs`. Two important things for better reproducing our results rely on the GPUs and the dataloader. The reproduction of our ImageNet experiments is not trivial. First, as the teacher-student model is used in our ImageNet experiments, it will occupy a lot of GPU memory. We trained on 3 NVIDIA A40 GPUs, each A40 has 48 GB of GPU memory, with a batch size of 170 per GPU for as much as ten days. Second, for faster training on ImageNet, we used NVIDIA DALI dataloader to fetch the data into GPUs for the image pre-processing. This dataloader could effect training as it uses a slightly different image resizing algorithm than the standard PyTorch dataloader. To keep results consistent with other methods, we do the inference with the standard PyTorch dataloader.

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

## A  APPENDIX

### A.1  CASCADED EMAS AS 2ND ORDER FILTER

Here we provide an alternative solution for expressing cascaded EMAs as 2nd order filter. We first express $g_i$ in time step $i$ as:

$$
\begin{aligned}
g_i &= (1-\alpha)g_{i-1} + \alpha m_i \\
&= (1-\alpha)g_{i-1} + \alpha\Big[(1-\gamma)m_{i-1} + \gamma\nabla_{\theta_i}\Big] \\
&= (1-\alpha)g_{i-1} + \alpha\gamma\nabla_{\theta_i} + (1-\gamma)\alpha m_{i-1}
\end{aligned}
\tag{19}
$$

In time step $i-1$, we denote $g_{i-1}$ as:

$$
g_{i-1} = (1-\alpha)g_{i-2} + \alpha m_{i-1}
\tag{20}
$$

where we have:

$$
\alpha m_{i-1} = g_{i-1} - (1-\alpha)g_{i-2}
\tag{21}
$$

Writing the $\alpha m_{i-1}$ in Eq. (21) into Eq. (19) we get:

$$
\begin{aligned}
g_i &= \alpha\gamma\nabla_{\theta_i} + (1-\alpha)g_{i-1} + (1-\gamma)\Big[g_{i-1} - (1-\alpha)g_{i-2}\Big] \\
&= \alpha\gamma\nabla_{\theta_i} - (\alpha+\gamma-2)g_{i-1} - (\alpha-1)(\gamma-1)g_{i-2}
\end{aligned}
\tag{22}
$$

