# OpenReview forum: "Understanding weight-magnitude hyperparameters in training binary networks"
_ICLR.cc/2023/Conference — ICLR 2023 poster_

### Official Review · Reviewer_H9io · 2022-10-13

**Confidence:** 3
**Correctness:** 3
**Technical Novelty And Significance:** 2
**Empirical Novelty And Significance:** 2
**Recommendation:** 5

**Clarity, Quality, Novelty And Reproducibility:**

Regarding the mathematical arguments in the paper, the clarity could be improved by using more formal language with theorems, assumptions, and so on explicitly stated. The current mathematical exposition is rather hand-wavy. One weakness is that the analysis might not extend to adaptive optimizers, could the authors comment on this?

The paper also repeatedly refers to “our optimizer”. Could you clarify what this optimizer is?

Regarding the experiments, the figures and experimental setup are rather clear. However, I find the hyperparameters to be rather strange. E.g. a batch size of 510 is used. Could the authors motivate their choice of hyperparameters? Furthermore, could the authors discuss how the choice of hyperparameters for their method and their competitors be discussed?

The empirical results are also broken down by one or two-stage training. Is there any reason why two-stage training might not be preferred? It seems to give better results, but maybe it consumes more resources? Furthermore, could your method be applied to two-stage training and then improve the final results?





**Strength And Weaknesses:**

Strengths:

* The authors provide a clear and intuitive explanation for how hyperparameters interact when training binary networks with SGD.
* The authors clearly verify their mathematical arguments experimentally.

Weaknesses:

* The mathematical arguments are rather simple and not surprising.
* It is not clear how the analysis extends to adaptive optimizers such as Adam. Note that Adam is used for the imagenet experiments.
* The authors refer to “our optimizer”, but it seems to me like this is just a way to parametrize well-known methods.
* Only results matching SOTA are reached.




**Summary Of The Paper:**

For binary neural networks, some hyperparameters and optimization tricks such as weight decay lose their meaning. In this paper, the authors reinterpret these methods through a perspective of gradient filtering. Specifically, if the binary weights are taken to be the sign of latent weights the authors show that SGD with momentum and weight decay is equal to smoothing the gradient twice with an EMA filter.




**Summary Of The Review:**

The paper visits an interesting topic and brings some relevant and thoughtful, albeit maybe simple, mathematical arguments which illuminate how hyperparameters interact when using SGD for binary networks. It is not clear how the analysis extends to adaptive optimizers, and what exactly the proposed method is is also not clear. While the authors clearly verify the theoretical results, the method does not seem to bring any performance benefits over two-stage methods.

---

> ### Author Response · Authors · 2022-11-18
> **Response**
>
> > The mathematical arguments are rather simple and not surprising.
>
> Thank you. It was not trivial to arrive at a simple exposition. The arguments might not be surprising, we do think that the consequences are, because it leads to a substantially different explanation for what happens during optimization. Namely filtering gradients instead of optimizing a latent weight.
>
> > It is not clear how the analysis extends to adaptive optimizers such as Adam
>
> > One weakness is that the analysis might not extend to adaptive optimizers, could the authors comment on this?
>
> Thank you for the opportunity. Adam extends momentum by scaling current gradients by past statistics. As such it does not depend on any weight-magnitude hyperparameters. Thus, we believe that adam, and other flavors of adaptive optimizers are orthogonal to our contribution and thus can freely be used without changing any of our results. We updated the last paragraph of Section 5: “Discussion and Limitations” of the updated PDF.
>
> >The paper also repeatedly refers to “our optimizer”. Could you clarify what this optimizer is?
>
> Yes!  “Our optimizer” is indeed not the best way to put it. The main difference is the latent weight vs the gradient filtering perspective. We do believe that we should call it a new optimizer to reflect the hyperparameters and the specific setting that is used. We updated the text to reflect this, as well as specifically mention the new optimizer in the summary of Section 3. Please see our updated PDF. (major changes marked in ‘red text’).
>
>
> > Regarding the experiments, the figures and experimental setup are rather clear. However, I find the hyperparameters to be rather strange. E.g. a batch size of 510 is used. Could the authors motivate their choice of hyperparameters?
>
> The batch size of 510 specifically for training ReActNet was chosen because the batches needed to be split between 3 GPUs to be able to fit everything in VRAM, so we picked a batch size of 170 per GPU to get close to the original batch size of 512.
>
> > Note that Adam is used for the imagenet experiments.
>
> For existing hyperparameters we followed previous works as close as possible, so this includes the batch size and hyperparameters related to optimizing the real-valued parameters in the networks. Since in the original paper the real-valued parameters in ReActNet are trained using Adam, we also use Adam with the same hyperparameters to optimize the real-valued parameters.
>
> > Furthermore, could the authors discuss how the choice of hyperparameters for their method and their competitors be discussed?
>
> Yes! For the hyperparameters of the filtering-based optimizer we can say that the $\alpha$ for the first filter approximately translates to weight decay times the learning rate for latent weights, so we also used this as a basis for tuning $\alpha$. For $\gamma$ we simply kept the same value of 0.1 as usually is used for momentum, because the second filter in our new method maps exactly to momentum.
>
> > The empirical results are also broken down by one or two-stage training. Is there any reason why two-stage training might not be preferred? It seems to give better results, but maybe it consumes more resources?
>
> Yes, we regret that we were not clear.  The main downside of using two-stage training is that it takes twice the amount of time and resources to optimize the networks. We, however, also believe that two-stage training introduces extra complexity in the training process which should be avoided if it can be. For example, perhaps the same improvements can be made by simply training for twice as long. We clarified the text, see the top of section 4.1 of the updated PDF.
>
> > Only results matching SOTA are reached.
>
> > Furthermore, could your method be applied to two-stage training and then improve the final results?
>
> Thank you for the suggestion. Indeed, by spending more effort on tuning we might be able to slightly  improve results. Yet, we were not very clear in the goal of the experiment. We merely aimed to show that our analysis is not worse. For experimental improvements we kindly refer the reviewer to subsection 4.2 of the new PDF.
>
> As a final remark, we hope that we now addressed the low scores on Technical Novelty and Empirical Novelty here in the rebuttal and also in the updated version of our PDF.

---

### Official Review · Reviewer_YST8 · 2022-10-14

**Confidence:** 3
**Correctness:** 3
**Technical Novelty And Significance:** 3
**Empirical Novelty And Significance:** 1
**Recommendation:** 8

**Clarity, Quality, Novelty And Reproducibility:**

The paper is clear and original. While the authors did not include code with their submission, they have stated that they intend to release their code to make their experiments reproducible.

**Strength And Weaknesses:**

Strengths:
- This paper builds upon the latent-weight free interpretation of binary neural networks, and provides a new understanding of traditional optimization hyperparameters in such a setting.

Weaknesses:
- This paper fails to demonstrate a significant empirical benefit from their new interpretation of binary neural network optimization beyond a reduction in the number of hyperparameters in some cases.

**Summary Of The Paper:**

This paper provides an analysis of standard optimization hyperparameters like weight initialization, learning rate, weight decay,  and momentum in the context of the latent-weight free interpretation of binary neural networks.

In a binary neural network, all weights and activations are +1 or -1. Although the weights are binary, they are typically optimized with real-valued gradients. In the latent-weight free interpretation, optimization can be viewed as a weight-flipping algorithm. When the gradients have accumulated past a certain threshold, the optimizer flips the weights.

But what should our understanding of traditional optimizer hyperparameters like weight initialization and learning rate be if we are to view optimization as weight-flipping? This paper argues that we should interpret them as impulse response filters on the gradients. This understanding reduces the number of hyperparameters that need to be tuned.

**Summary Of The Review:**

This paper provides an interesting new understanding of optimization in the binary neural network setting. However, it also fails to show significant empirical benefits from such an understanding.

I am on the fence about recommending this paper for acceptance, because it's not clear to me how the community can use this newfound understanding to significantly advance our theoretical understanding of binary neural networks or their empirical application. However, I also have low confidence in my assessment. I think the authors can improve the paper by adding new empirical or theoretical results, or at least sketching out how their contribution is significant to the community.

Edit: Thanks for the authors' response. The quotes from relevant papers in the literature has increased my confidence in the significance of the work. Even though the empirical contributions are still weak, I see the great value this piece of work brings in filling this gap in the literature. I have updated my score to reflect that. I highly encourage the authors to re-write the Introduction section to emphasize the significance of this work by highlight the gaps in the literature and how this work fills them.

---

> ### Author Response · Authors · 2022-11-18
> **Response**
>
> > I think the authors can improve the paper by adding new empirical or theoretical results
>
> > This paper fails to demonstrate a significant empirical benefit from their new interpretation of binary neural network optimization beyond a reduction in the number of hyperparameters in some cases.
>
> Agreed. We did improve on this by adding a new sub-section on the empirical benefits of reducing the hyperparameters. Fewer hyperparameters require much less computation for Bayesian hyperparameter search. Please see our new section 4.2 of the updated PDF. (major changes marked in ‘red text’).
>
>
> > because it's not clear to me how the community can use this newfound understanding to significantly advance our theoretical understanding of binary neural networks or their empirical application.
> We believe that the future of BNN optimization lies in binary specific optimizers (BOP, AdamBNN, etc) and understanding their hyperparameters is a crucial first step towards a better theoretical understanding of binary neural networks.
>
> > or at least sketching out how their contribution is significant to the community.
>
> The community is currently struggling with these hyperparameters. Several works from the community remark on the lack of their understanding (Please see our answer to Reviewer AvuJ, and the first paragraph of our “related work” section), and we believe that our paper addresses this research gap which is self-identified by the community.
>
> In summary, we hope that we could address the low scores on Technical Novelty and Empirical Novelty here in the rebuttal and also in the updated version of our PDF.

---

### Official Review · Reviewer_AvuJ · 2022-10-24

**Confidence:** 2
**Correctness:** 3
**Technical Novelty And Significance:** 3
**Empirical Novelty And Significance:** 3
**Recommendation:** 6

**Clarity, Quality, Novelty And Reproducibility:**

- Figure 1 seems pretty unnecessary. I imagine most readers will be familiar with with binary vs real valued numbers.
- The analysis is quite simple. This is a good thing, but I don't know this area well enough to comment on how novel it is.

**Strength And Weaknesses:**

**Strengths**
- The main idea of the paper is simple.
- Removing hyperparameters and reducing the need for tuning is important---especially in architectures that are large or expensive to train.
- The paper is easy to follow.

**Weaknesses**
- Experiments are limited to two models and datasets, both of which are image classification.

**Summary Of The Paper:**

Binary Neural Networks use values of -1 and +1 for the network's weights instead of floating point number to save on memory. Successful training of these networks, and their ultimate performance, depends on several hyperparameters. This paper redefines or removes many of these hyperparameters. This means that there are fewer hyperparameters to tune and removes the need for latent weights. Experiments support that there is little loss in performance compared to methods with latent weights.

**Summary Of The Review:**

The paper is clear and presents a nice idea. The experiments support the main claim of the paper but are limited in scope. I do not know the literature on Binary Neural Networks well enough to comment on the paper's novelty.

---

> ### Author Response · Authors · 2022-11-18
> **Response**
>
> > Experiments are limited to two models and datasets, both of which are image classification.
>
> Yes! We added an additional model and a different modality (audio). Kindly refer to section 4.2 of the updated PDF. (major changes marked in ‘red text’).
>
> > Figure 1 seems pretty unnecessary. I imagine most readers will be familiar with with binary vs real valued numbers.
>
> Yes, this is probably true. I hope the reviewer does not penalize us for keeping the figure, as it makes our paper easier to understand.
>
> > Experiments support that there is little loss in performance compared to methods with latent weights.
>
> We are indeed on-par with respect to performance. We do, however, have a clear benefit in the number of hyperparameters to tune. We show the computational benefits in Section 4.2 of the updated PDF.
>
> > I do not know the literature on Binary Neural Networks well enough to comment on the paper's novelty.
> > The analysis is quite simple. This is a good thing, but I don't know this area well enough to comment on how novel it is.
>
> Thank you! It took quite some effort to arrive at this perceived simplicity. With respect to the novelty: It is novel in the sense that we are not aware of any other paper doing this analysis.
> Note, however, that there are many papers that remark on the unexplained interpretation of these hyperparameters, as exemplified by the remarks of the following papers (which we all cite):
> - Liu et al. (2021)
>    - “However, for a binary neural network, the effect of weight decay is less straightforward”
> - Helwegen et al. (2019)
>   - “scaling of the learning rate does not have the role one may expect”
>   - “As we have argued, it is not clear that applying L2-regularization or weight decay to the latent weights should lead to any regularization at all.”
> - Tang et al. (2017)
>   - “However, the L2 regularization is contradictory to what a binary network wants to achieve.”
> - Martinez et al. (2020)
>   - “Note that weights at stage 2 are either 1 or −1, so applying an L2 regularization term to them does not make sense.”
>
> Thus, there is a clear gap in the research field about the understanding of these hyperparameters. Our paper fills this research gap.

---

### Official Review · Reviewer_gyrn · 2022-10-27

**Confidence:** 3
**Correctness:** 3
**Technical Novelty And Significance:** 2
**Empirical Novelty And Significance:** 2
**Recommendation:** 6

**Clarity, Quality, Novelty And Reproducibility:**

The paper well clarifies the interpretability issue of binary BNN weights and mag-based hyperparameters. The writing is good and easy to follow; yet the experiment is expected to be strengthened.

The paper does not provide code and pseudo code/algorithm flow.


**Strength And Weaknesses:**

Pros:
- This paper has good motivation for interpreting the magnitude-based hyperparameters on BNN.
- It provides some interesting insights such as:
  - A gradient filtering perspective on latent weight hyperparameters.
  - A justification of which magnitude-based hyperparameters to use and offer a setting with fewer hyperparameters to tune.

Cons:
- It would be better to add a summary paragraph in section 3 to state their concise conclusion of magnitude-based hyperparameters.
- As mentioned, the performance improvement is marginal in Table 2 based on ReActNet-A.
- The paper claimed that it provided a “simplified” setting with fewer hyperparameters to tune. However, I did not see the big advantages of the proposed optimizer compared to latent-weight optimization in terms of accuracy, and training/optimization speed.
- The experiment section is relatively weak as the experiment is given on one architecture based on a few baseline methods, “ReActNet-A” on Imagenet and “Bi-Real” on CIFAR. It is expected to show how the proposed optimizer performs on several recent BNN optimization methods with various types of architecture and different types of datasets (modality such as visual and text, distribution such as natural, digital, fine-grained) and task (classification, detection, and segmentation.)
- The metrics in Table 2 haven't been explained well. More details should be given.


**Summary Of The Paper:**

This paper focuses on the analysis of hyperparameters in binary neural networks (BNN). The hyperparameters could explain the BNNs with latent real-valued weights during training. However, the magnitude of binary weights is not interpretable with these hyperparameters yet. This paper provides an interpretation of these magnitude-based hyperparameters based on higher-order gradient filtering during network optimization.

**Summary Of The Review:**

Overall, this paper provides some interesting insights for interpreting binary weights of BNN to magnitude-based hyperparameters. However, the major concern arises in the lack of experiments to support their statements. Also, the proposed optimizer does not offer many advantages compared with the latent-weight-based optimization.

**Post after rebuttal**

I appreciate the additional experimental results and detailed responses provided by the authors. After checking the latest revision in the manuscript, I decided to raise my score to encourage further exploration of hyperparameters for binary neural networks.

---

> ### Author Response · Authors · 2022-11-18
> **Response**
>
> > It would be better to add a summary paragraph in section 3 to state their concise conclusion of magnitude-based hyperparameters.
>
> Excellent suggestion. We added a summary, and a new table (Table 1). Please see the updated PDF (major changes marked in ‘red text’).
>
> > As mentioned, the performance improvement is marginal in Table 2 based on ReActNet-A.
>
> This is true. Our goal is not to be better, but only to validate that our simplified setting is equivalent. We clarified the text to better reflect this (first paragraph of Section 4.1).
>
> > - yet the experiment is expected to be strengthened.
> > - However, the major concern arises in the lack of experiments to support their statements.
> > - The paper claimed that it provided a “simplified” setting with fewer hyperparameters to tune. However, I did not see the big advantages of the proposed optimizer compared to latent-weight optimization in terms of accuracy, and training/optimization speed.
>
> We agree. To make the empirical computational advantage of the reduction of hyperparameters clearer we now show this empirically in a hyperparameter tuning experiment for a real-world scenario when the hyperparameters are unknown, see section 4.2 of the updated PDF.
>
>
> >  It is expected to show how the proposed optimizer performs on several recent BNN optimization methods with various types of architecture and different types of datasets (modality such as visual and text, distribution such as natural, digital, fine-grained) and task (classification, detection, and segmentation.)
>
> Agreed. Please see the updated section 4.2, where we added a new architecture, and a new modality.
>
> > The metrics in Table 2 haven't been explained well. More details should be given.
>
> We find it unfortunate that we created this confusion. The metrics are about runtime efficiency. We took these metrics from the Re-ActNet paper, yet, for our goals they are not relevant. Our filtering viewpoint is slightly more efficient because we have no scaling factors, but this is not the main point of our paper. To remove the confusion we removed these metrics from the table.
>
> > The paper does not provide code and pseudo code/algorithm flow.
>
> We will make source code available.
>
>
> To conclude, we hope that the reviewer agrees that the raised issues on Technical Novelty and Empirical Novelty are now better addressed here, and in the updated version of our PDF.

---

### Comment · Area_Chair_qYPS · 2022-12-01
**Discussion**

I definitely like the spirit of the paper's contribution. It aims at simplifying and clarifying existing practices (which grow with new and new heuristics) for training binary neural networks by analysis. In this regard it stands out in the "binary NN" field, which is mostly empirical at present.

I have the following comments and concerns which I invite everybody to discuss (authors, please limit to one comment).

### Soundness
The authors should take into account that their research is about currently successful but heuristic and unclear methods. I.e. the work is not based on sound research. Both SGD with latent weights and BOP have no clear relation to optimization. Their equivalence, which is taken one step further in this paper, does not help with this issue, reinterpreting missing latent weights as accumulated gradients does not make it more sound. In particular, it does not justify by itself the 2nd order filtering design. It would be appropriate therefore to show that 2nd order filtering is useful theoretically and practically. E.g. noticing that SGD with Nesterov momentum is also a second order filtering. The departing point of the work is the invariance of the network to scaling of latent weights, known as "latent weights do not exist" problem. However, the straight-through gradient propagation for binary activations is used at the same time and nobody seems to be concerned with that the "activation scales do not exist" problem. Clearly, gradient filtering explanation cannot be applied to activations. So the approach cannot resolve the whole issue. Also, suppose the weights are quantized to 2 bits, what's then, do the latent weights exist or not?

The paper shows a reduction in the number of hyperparameters and their new interpretation starting from BNN setup and analyzing it. However, it is not granted that the excessive number of hyperparameters in the BNN setup (1-5) plus *separate* learning hyperparameters for continuous parameters are really needed in the first place.

### Related work

The work should discuss alternative approaches for making the binary training methods more sound. One such direction is considering stochastic relaxation of discrete optimization, where the optimization problem is well posed and the gradient estimators and training algorithms can be rationally designed. Some estimators and algorithm would be completely different from the studied in the paper, e.g. Peters and Welling "Probabilistic Binary Neural Networks". However there are also estimators and methods very similar to the ST+SGD heuristics, in particular see

Shekhovtsov and Yanush (2021): "Reintroducing Straight-Through Estimators as Principled Methods for Stochastic Binary Networks",
Sec. 3 "Latent Weights do Exist!" and

Meng et al. (2020): "Training binary neural networks using the Bayesian learning rule", Sec. 3.1. "BayesBiNN optimizer",

both have a sound concept of latent weights and latent weight decay, implementing variational Bayesian learning.
In quantization, a noisy quantization model is proposed by Louizos et al. (2018), "Relaxed Quantization for Discretized Neural Networks"
and specializes in the binary case to a similar treatment of weights and activations to the above.

These approaches successfully resolve the above-discussed lack of soundness issues. In particular the same formalism works for weights and activations, for quantized and binary. In particular, the scale ambiguity of latent weights does not arise. However they are more complex and have not been demonstrated to outperform the mainstream heuristic techniques.

### Technical remarks

1) The Xavier / He initializations play two roles: first, keeping the variance of pre-activations at initialization close to 1, which can be alternatively (and better) achieved by BN, and second, determining the relative local learning rate of weights versus e.g. biases, and other (possibly real-valued) parameters in the network. It is well known that rescaling weights, even when preserving the forward pass, can have a significant impact for the training, see Neyshabur et al. (2015) Path-SGD: Path-Normalized Optimization in Deep Neural Networks.

2) Perhaps update (18) could be more conveniently expressed and better interpreted using only $g_{t-1}$ and the past two gradients.

3) "not having enough time to converge to a good result in the second half" -- this seems to be a flaw of the experimental setup not to allow sufficient time for convergence.

4)  I disagree that introducing $a = \epsilon \lambda$  reduces the number of hyperparameters, because the learning rate $\epsilon$ is the global learning rate which I assume is still needed for learning other continuous parameters (biases, BN parameters, skip branches).

### Minor

* FF ratio appears to be the average flip frequency, i.e. not a ratio of two dynamic quantities, therefore the term is not very appropriate.
* In Fig 4 there is no (a-c).
* Language and paper layout need to be polished.

---

> ### Author Response · Authors · 2022-12-09
> **response**
>
> # Technical remarks
>
> 1. > The Xavier / He initializations play two roles: first, keeping the variance of pre-activations at initialization close to 1, which can be alternatively (and better) achieved by BN, and second, determining the relative local learning rate of weights versus e.g. biases, and other (possibly real-valued) parameters in the network. It is well known that rescaling weights, even when preserving the forward pass, can have a significant impact for the training, see Neyshabur et al. (2015) Path-SGD: Path-Normalized Optimization in Deep Neural Networks.
>
>   We agree with this observation. But we are a bit confused about which part of the paper should be updated. Could you kindly elaborate and help us in our confusion?
>
> 2. > Perhaps update (18) could be more conveniently expressed and better interpreted using only $g_{t-1}$ and the past two gradients.
>
> As far as we know, this is not possible without changing the working of the filter. A filter that looks back at the last two inputs and the last output, simply functions differently from a filter that looks at the last input and the last two outputs.
>
> 3. > "not having enough time to converge to a good result in the second half" -- this seems to be a flaw of the experimental setup not to allow sufficient time for convergence.
>
> Yes, this should have been better worded. What we meant by this is that as long as the FF ratio is too high, the network does not seem to learn anything. Since this happens for around 75% of the training time for alpha=1e-6 the network only has 25% of the time to learn and converge to a good solution. This could indeed be fixed by either increasing the amount of epochs, however the learning would then still only happen at the last 25% percent, which is very inefficient or $\alpha$ could be decayed more quickly at the start, however that would just be equivalent to starting with a lower $\alpha$. To make this more clear we’ve updated the caption as follows:
> ‘’’Figure 5: FF ratio and accuracy for varying alphas. BNNs are sensitive to α, similar to how they are sensitive to weight decay, because α strongly influences the FF ratio, which is tightly coupled to accuracy. An initially too small FF ratio can lead to quick convergence during the start of training, but stagnation in the second half. While a large FF ratio leads to noisier training and lower accuracy at the start, but can catch up in the second part when the FF ratio starts decreasing because of α decay. If the initial FF ratio is too large, however, then learning starts very late.’’’
>
> 4. > I disagree that introducing $a=\epsilon\lambda$ reduces the number of hyperparameters, because the learning rate $\epsilon$ is the global learning rate which I assume is still needed for learning other continuous parameters (biases, BN parameters, skip branches).
>
> Indeed, if you start from a setting where all of the hyperparameter values are shared between real-valued and binary parameters it seems like using a separate optimizer for binary parameters will always introduce more hyperparameters compared to using the same optimizer with the same HP for both. However, we believe that in the setting where we start from, the hyperparameters have such different effects on real-valued parameters and binary parameters, it seems like an arbitrary constraint to keep them equal between the two. So this is why we are arguing that combining the learning rate and weight decay into $\alpha$ reduces the number of hyperparameters. We do agree that we should make this more clear in the paper itself.
>
> # Minor
> > FF ratio appears to be the average flip frequency, i.e. not a ratio of two dynamic quantities, therefore the term is not very appropriate.
>
> Yes, it can indeed be interpreted as the average frequency with which the weights flip, which is similar to the ratio of all weights that flip at each time step. Yet, we are kind of stuck with this term, as it was introduced by others and it might introduce confusion if we introduce other terminology. Thus, if OK, we prefer to keep this term to stay consistent with the previous literature.
>
> > In Fig 4 there is no (a-c).
>
> Excuse us, (a-c) refers to the different lines in the Figure, we updated the legend
>
> > Language and paper layout need to be polished
>
> Agreed. With the new paragraphs, we now polished the layout and the language making it better readable and presentable.

---

> ### Author Response · Authors · 2022-12-09
> **response**
>
> Thank you for these insightful and thoughtful comments. Adding these insights to the paper will strengthen the work, and we accordingly added some new paragraphs. We are no longer able to upload a new version of the PDF, but we will show the added text below.
>
> # Soundness
>
> Yes, our work builds on empirically motivated approaches which are not all very well theoretically motivated (for example using straight-through gradients) and as such we do not improve the soundness of these empirical methods themselves. There is a dichotomy between popular work with strong, but unsound, empirical results on the one hand, versus, on the other hand, more theoretically sound work that might be more complex and yielding less strong empirical results. To have a discussion about this is valuable not only to the paper, but to the whole field. Thus, we added the following paragraph to the discussion section:
>
> > One last point we should touch upon is soundness. Even though our filtering perspective provides a better explanation to hyperparameters, it does not explain why empirically optimizing BNNs with stochastic gradient descent (SGD) works as well as it does. Whereas SGD has extensive theoretical background for real-valued networks, such sound theory does not exist for currently popular empirical BNN methods. Better understanding BNN optimization is key future research where our work takes a small step towards that objective.
>
>
> # Related Work
>
> Absolutely! The suggested citations are clearly relevant for discussing sound vs unsound BNN approaches. We added the following paragraph to the related work:
>
>
> > ***Sound optimization approaches***. Instead of using heuristics to approximate gradient descent on discrete binary values, several works take a more principled approach. Peters & Welling (2018) proposes a probabilistic training method for BNN with the stochastic relaxation of discrete optimization. Shekhovtsov & Yanush (2021) present a theoretical understanding of straight through estimators (STE) (Bengio et al., 2013) for stochastic binary networks (SBNs), and show that, for SBNs, latent weights can be formally defined and several empirical update rules are related to sound optimization schemes such as variational Bayesian learning. Similarly, Meng et al. (2020) propose a Bayesian perspective by relaxing the discrete optimization using a distribution over the binary variable. Instead of specifying a
> Bernoulli distribution over the weights, Louizos et al. (2018) formulate a noisy quantizer for general quantization with using a relaxation procedure for enabling gradient-based optimization. Even though these approaches provide more theoretical justification in optimizing BNNs, they are more complex with either relying on stochastic settings or discrete relaxation training procedures. Moreover, these methods do not (yet) empirically reach a similar accuracy as current mainstream heuristic methods such as Liu et al. (2018), and Liu et al. (2020). In our paper, we build on the mainstream approaches, to get good empirical results, but improve the understanding of their properties, taking a step towards better theoretical understanding of empirical approaches.
>
>
> ***Reference***
>
> [Peters & Welling, 2018] Probabilistic Binary Neural Networks.
>
> [Shekhovtsov & Yanush (2021)] Reintroducing Straight-Through Estimators as Principled Methods for Stochastic Binary Networks.
>
> [Bengio et al. (2013)] Estimating or propagating gradients through stochastic neurons for conditional computation.
>
> [Meng et al. (2020)] Training binary neural networks using the Bayesian learning rule.
>
> [Louizos et al. (2018)] Relaxed Quantization for Discretized Neural Networks.
>
> [Liu et al. (2018)] Bi-real net: Enhancing the performance of 1-bit cnns with improved representational capability and advanced training algorithm
>
> [Liu et al. (2020)] Reactnet: Towards precise binary neural network with generalized activation functions.

---

### Decision · Program_Chairs · 2023-01-20

**Decision:**

Accept: poster

**Justification For Why Not Higher Score:**

The contribution of the paper was evaluated as not very strong and the paper is not as comprehensive in terms of analysis, arguments and connections to the literature. The practical gain was recognized by reviewers as useful but not exciting.

**Justification For Why Not Lower Score:**

The paper contributes a useful knowledge, clearly demonstrated theoretically and experimentally. The average score is positive. No major issues identified.

**Metareview: Summary, Strengths And Weaknesses:**

### Description

The paper proposes an analysis of heuristic straight-through SGD methods for training binary neural networks. The current heuristics contain many hyperparameters such as latent weight initialization, latent weight learning rate, latent weight decay. Exploiting the invariances of the learning algorithm due to the binarization operation, the paper infers a simplified equivalent learning algorithm with effectively fewer hyperparameters and a new interpretation as second order gradient filtering.

### Summary of Evaluation
The weaknesses pointed out by reviewers were:
* The mathematical arguments are rather simple and not surprising (H9io).
* It is not clear how the analysis extends to adaptive optimizers such as Adam. Note that Adam is used for the imagenet experiments (H9io).
* Several reviewers remarked they do not see an advantage in terms of accuracy.
* Several reviewers and AC pointed out minor issues (clarity, literature, technical)

I believe all these issues have been clarified. In particular the Adam is used for optimizing real-valued parameters along the considered simplified optimizer for binary weights. And in particular, the method is not supposed to improve accuracy. The paper was overall positively evaluated by reviewers and no issues requiring a major revision were identified. Despite the technical contribution is rather simple compared to many other papers at ICLR, it is useful and rather solid compared to many papers in the binary NNs field and seems to offer a new view allowing to build new connections. Therefore we give it a green light.

### Recommendations for the final version

Authors are encouraged to make improvements for the final version:
* Discuss related work on interpreting latent weights (as already discussed).
* Show that the gradient filter is really estimating the gradient: assuming that $\nabla L_i$ are i.i.d. for all $i$ (ie. it is a stochastic gradient evaluated at the same point in the parameters space), show that the filtered $g_i$ (18) is unbiased: $E[g_i] = E[\nabla L_i]$. Adjust scaling to satisfy this, if necessary. Visualize filter weights in comparison with EWA (see below)
* Look up and connect to the related work on applying other filters than EWA in deep learning training,
e.g. Yang "Stochastic Gradient Variance Reduction by Solving a Filtering Problem"
* Improve the paper clarity, conciseness and soundness of arguments

### Follow-up on the discussion

I thank the authors for the explanations.

>  The Xavier / He initializations

This was concerning the paragraph "Latent weight initialization" where the paper says " no longer makes sense to use real-valued weight initialization techniques". I am quite convinced that the gradient filtering view allows to simplify the existing heuristic SGD algorithm from which the paper starts. However I am not so convinced that this allows to fully change the interpretation. In particular more argumentation of the claim "no longer makes sense to use real-valued weight initialization techniques" is neede. Perhaps saying that the variance analysis underlying these techniques does not apply would be verifiable and more understandable. The random initialization at zero is still used in the proposed scheme, somewhat artificially. Also note that in the Adam they perform a bias correction of the EWA averages for the first and second moment, which assumes zero initialization. The correction is such that the filtered mini-batch gradient is an unbiased estimate of the full gradient (assuming we do not make optimization steps, ie. the learning rate is infinitesimally small).

> "Expand using past two gradients" -- As far as we know, this is not possible without changing the working of the filter.

Yes, I agree.
Another, perhaps a better option to understand it would be to plot the filter weights. I.e. to expand (18) as a linear combination of all past gradients only and show the weights of this linear combination as plot. E.g. with EWA the weights can be written explicitly $w_0 = (1-q)^n$, $w_i = (1-q)^{n-i}q$. It should be possible to compute such weights for the proposed filter at least numerically and visualize for different choices of a parameters. This would be very complementary to Fig 2. Consider improving the paper. Then, one could attempt to make a rational choice of the filter. I would argue that a filter that makes a flat average of $k$ past gradients would be useful -- it has a good variance reduction and low lag, however implementing it requires remembering $k$ past gradients. If the second order could move the weight mass more towards such a rectangular filter compared to EWA that would be convincing for me and possibly useful in a wider context. Please check and discuss in the paper the related work on gradient filtering. It would strengthen the paper to show such connections.







**Note From Pc:**

if the above contains the word "oral" or "spotlight" please see: "oral" presentation means -> notable-top-5% and "spotlight" means -> notable-top-25%. As stated in our emails, we are disassociating presentation type from AC recommendations

**Summary Of Ac-Reviewer Meeting:**

It was not possible to find an overlap in reviewers and AC schedule to have a meeting with a reasonable quorum.